# UNIFIED ANOMALY DETECTION VIA MULTI-SCALE CONTRASTED MEMORY

## ABSTRACT

Deep anomaly detection (AD) aims to provide robust and efficient classifiers for one-class and imbalanced outlier-exposure settings. However current models still struggle on edge-case normal samples and are often unable to keep high performance over different scales of anomalies. Additionally, there is a lack of a unified framework that efficiently addresses both OC and OE settings. To address these limitations, we present a novel two-stage method which leverages multi-scale normal prototypes during training to compute an anomaly deviation score. First, we employ a novel memory-augmented contrastive learning (CL) to jointly learn representations and memory modules across multiple scales. This allows us to effectively capture subtle features of normal data while adapting to varying levels of anomaly complexity. Then, we train an efficient anomaly distance-based detector that computes spatial deviation maps between the learned prototypes and incoming observations. Our model outperforms the SoTA on a wide range of anomalies, including object, style, and local anomalies, as well as face presentation attacks, while being on par with SoTa out-of-distribution detectors. Notably, it stands as the first model capable of maintaining exceptional performance across both settings.

## 1 INTRODUCTION

Detecting deviations from a well-defined normal baseline is a central challenge in modern machine learning. Anomaly detection (AD) differs significantly from conventional binary classification, due to the intricate, incomplete, and ill-defined nature of anomalies. This has led to the emergence of deep AD methods that provide greater stability with imbalanced training dataset. However, existing AD models still have some limitations. **(1)** There is a hard trade-off between remembering edge-case normal samples and remaining generalizable enough toward anomalies. This *lack of normal sample memorization* often leads to high false reject rates on harder samples. **(2)** These models tend to focus on either local low-scale anomalies or global object-oriented anomalies but fail to combine both. Current models often remain highly dataset-dependent and *do not explicitly use multi-scaling*. **(3)** AD ***lacks an efficient unified framework*** which could easily tackle OC and imbalanced OE detection. Indeed, existing methods are either introduced as OC or OE detectors, with different specialized approaches and set of hyper-parameters (as can be seen in Fig. 1), a noteworthy limitation persists. Current methods are confined to their original design setting, necessitating a complete overhaul of the algorithm when transitioning from scenarios lacking anomaly samples to those enriched with them. This impracticality becomes evident in real-world applications where initially only normal samples are available, and the acquisition of anomaly samples occurs later.

To address these limitations, we introduce a novel two-stage AD model named AnoMem which memorizes during training multi-scale normal class prototypes to compute an anomaly deviation score at several scales. Unlike previous memory bank equiped methods (Gong et al., 2019; Park et al., 2020), our normal memory layers encompass multiple scales, enhancing both AD and the quality of learned representations. Additionally, by using the modern and backpropagable Hopfield layers for memorization, our method is much more efficient

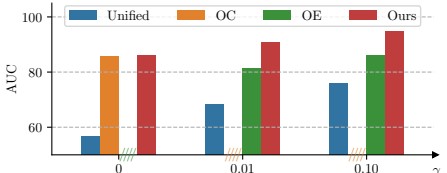

Figure 1: AnoMem vs. SoTA on CIFAR-100 with different anomaly ratio $\gamma$.

than nearest neighbor anomaly detectors (Bergman et al., 2020; Gui, 2021): those mentioned detec-

tors require keeping the *whole* normal set, while ours can learn the most representative samples with a fixed size. Through extensive experiments, we demonstrate that our method significantly outperforms all previous memory-equipped anomaly detectors. Our main contributions are the following:

- We propose to integrate memory modules into CL to remember normal class prototypes during training. In the first stage, we *jointly* learn representations and memory modules using CL, allowing for effective **normal sample memorization**. In the second stage, we learn to detect anomalies using the memorized prototypes. When anomalous samples are available, we train a detector on the spatial deviation maps between prototypes and observations. To our best knowledge, AnoMem is the first working well in both OC and OE [1] settings with a few anomalies, making it a **unifying method**. Our unified setting differs from existing ones. In semi-supervised AD, not all training samples are annotated, while open-set supervised AD (Ding et al., 2022) focuses on detecting unseen anomalies, which remains supervised with several anomaly types seen during training.
- AnoMem is further improved by using multi-scale normal prototypes for representation learning and AD. We introduce a novel way to efficiently memorize 2D features maps spatially. This enables our model to accurately detect low-scale, texture-oriented anomalies and higher-scale, object-oriented anomalies (**multi-scale AD**).
- We validate the efficiency of our method and compare it with SoTA methods on one-vs-all, out-of-distribution (OOD) and face anti-spoofing detection. We improve detection with up to 50% error relative improvement on object anomalies and 14% on face anti-spoofing.

## 2 RELATED WORK

**Memory Modules.** A memory module should achieve two main operations: (i) writing inside a memory from a given set of samples, and (ii) recovering from a partial or corrupted input the most similar sample in its memory with minimal error. Memory modules will differ on the amount of images they can memorize given a model size and the average reconstruction error.

A simple memory module is the nearest neighbor queue. Given a maximum size $M$, it stores the last $M$ samples representations in the queue. To remember an incomplete input $x$, it retrieves the nearest neighbor from the queue. A more effective memory module is the modern Hopfield layer (Ramsauer et al., 2021). It represents the memory as a learnable matrix of weights $X \in \mathbb{R}^{d \times N_{\text{Mem}}}$ and retrieves samples by recursively applying the following formula until convergence: $\xi^{(t+1)} = \text{softmax}\left(\beta \xi^{(t)} X\right) X^T$, where $\xi^{(0)}$ is the query vector and $\beta$ is the inverse temperature. Its form is similar to the attention mechanism in transformers, except it reapplies the self-attention until convergence. This layer has a very high memory capacity and remember samples with very low redundancy (Ramsauer et al., 2021). Subsequently, we call this layer a Hopfield layer of size $N_{\text{Mem}}$.

**Anomaly Detection.** AD involves classifying data into two categories: normal and anomaly. While the normal class is well-defined, the anomaly class encompasses all other variations, making it broader and more complex. This leads to a natural data imbalance. In the literature, various methods have been proposed, categorized into three families of approaches: pre-trained feature adaptation, discriminative, and generative methods (as outlined in (Salehi et al., 2022)). This paper categorizes existing methods based on the availability of anomalous samples, leading to two distinct categories: one-class AD (OC-AD) which exclusively use normal samples for training, and outlier-exposure AD (OE-AD) which incorporate a small set of anomalies during training to enhance adaptability.

There exist different **OC-AD** approaches. *Pretext task methods* learn to solve a different auxiliary task on normal data (Hendrycks et al., 2019; Jezequel et al., 2022b; Bergman & Hoshen, 2020), and the anomaly score reflects performance on this auxiliary task. *Two-stage methods* consist of representation learning and anomaly score estimation. After training an encoder on normal data using self-supervised learning (Cho et al., 2021; Tack et al., 2020; Chen et al., 2021; Zbontar et al., 2021) or an encoder pre-trained on additional datasets (Xiao et al., 2021; Reiss & Hoshen, 2023; Reiss et al., 2021), an OC classifier computes the anomaly score in the latent space (Sehwag et al., 2021; Li et al., 2021; Sohn et al., 2021). Some methods (Bergman et al., 2020; Gui, 2021) use a nearest neighbor queue for anomaly scoring. *Density estimation methods* estimates the normal distribution using deep density estimators like normalizing-flows (Kumar et al., 2021), variational models (Daniel et al., 2019) or diffusion models (Wyatt et al., 2022; Mirzaei et al., 2023). *Reconstruction methods* measure the reconstruction error of a bottleneck encoder-decoder, trained with denoising autoencoders

---

[1] We embrace the OC term, widely employed in OOD detection, which involves using external datasets as pseudo OOD samples. Our OE-AD refers to training data that includes both inlier and outlier samples.

(Perera et al., 2019; Schneider et al., 2022) or two-way GANs (Tuluptceva et al., 2019; Schlegl et al., 2017; Liu et al., 2021). Some incorporate memory in the latent space of an auto-encoder (Gong et al., 2019; Park et al., 2020) during training for optimal reconstruction. More recently, *knowledge distillation methods* have been adapted to AD by using the representation discrepancy of anomalies in the teacher-student model (Deng & Li, 2022; Cohen & Avidan, 2022).

**OE-AD** mainly revolves around two-stage methods and anomaly distance methods. We note however that some recent work has tried to generalize pretext task to OE-AD (Jezequel et al., 2022b). In the *OE-AD two-stage methods*, a supervised classifier with the anomalous samples is trained in the second stage instead of the aforementioned one-class estimator (Han et al., 2021). *Distance methods* directly use a distance to a centroid as the anomaly score and learn the model to maximize the anomaly distance on anomalous samples and minimize it on normal samples (Ruff et al., 2020; Jezequel et al., 2022a). To the best of our knowledge, **no OE-AD methods** in the literature **use any kind of memory mechanism** for the anomaly score computation. For instance, MemSeg (Yang et al., 2023) and PatchCore (Roth et al., 2022) use frozen pretrained encoder and memory is not learned. Moreover, there is no anomaly distance learning (only a fixed $L_2$ norm is used).

There is a closely related task of **anomaly localization (AL)**, which goal is to produce an anomaly heatmap. AL datasets range from defect localization (Bergmann et al., 2019) to surveillance video abnormal event detection (Mahadevan et al., 2010). Specialized methods targeting localization (Yi & Yoon, 2020; Li et al., 2021) have been introduced to efficiently solve this task.

Finally, the related task of **out-of-distribution (OOD)** detection aims to detect whether or not an observation has been sampled from the same distribution as the training set (Yu et al., 2023; Lafon et al., 2023; Djurisic et al., 2023). Compared to AD, the normal class of OOD will be highly multimodal since it can cover multiple semantic classes. Moreover, OOD will put more weight than in AD on the acquisition process of data (e.g., capture sensor, lighting, background). As a consequence, it is generally preferred for OOD detectors not to generalize too much on normal data, while it is the opposite in AD. In this paper, we present a ***generic unified method that jointly performs well on AD, OOD detection and face presentation attack detection***.

**Contrastive Learning (CL).** CL is a self-supervised representation learning method. It operates on the basis of two principles: **(1)** two different images should yield dissimilar representations, and **(2)** two different views of the same image should yield similar representations. The views of an image are characterized by a set of transformations $\mathcal{T}$. There have been many methods enforcing these two principles: SimCLR (Chen et al., 2020), Barlow-Twins (Zbontar et al., 2021) and VICReg (Bardes et al., 2022) with a Siamese network, MoCo (He et al., 2020) with a negative sample bank, BYOL (Grill et al., 2020) and SimSiam (Chen & He, 2021) with a teacher-student network or SwAV (Caron et al., 2020) with contrastive clusters. While some contrastive methods such as SimCLR and MoCo require negatives samples, other such as BYOL, SimSiam and SwAV do not.

In the simplest formulation, the pairs of representations to contrast are only considered from two views of a batch augmented by transformations $(t, t')$. In **SimCLR**, the following loss is minimized:$\mathcal{L}_{\text{CO}} = \frac{1}{2B} \sum_{k=1}^{B} \ell_{\text{NTX}}(z_k, z'_k) + \ell_{\text{NTX}}(z'_k, z_k)$ where $z, z'$ are the last features of the two augmented batches and $\ell_{\text{NTX}}$ is the Normalized Temperature-scaled Cross Entropy Loss (**NT-Xent**). In practice, minimizing $\mathcal{L}_{\text{CO}}$ will yield representations with the most angular spread variance, while retaining angular invariance in regard to $\mathcal{T}$. Memory mechanisms can also be used into CL as in (Dwibedi et al., 2021; Koohpayegani et al., 2021). During training, the positive and negative pairs are augmented with the samples nearest neighbors from a memory queue. This allows the model to reach better performance for smaller batch sizes.

## 3 PROPOSED METHOD

**Notation.** Let $\mathcal{X} = \{(x_k, y_k)\}_k$ be a training dataset with normal samples ($y_k = 1$) and potentially anomalies ($y_k = 0$). $f : \mathbb{R}^{H \times W \times C} \to \mathbb{R}^D$ is a backbone made of several stages $f^{(1)}, \cdots, f^{(S)}$ where the dimensions of the $s^{\text{th}}$ scale feature map are $H^{(s)} \times W^{(s)} \times C^{(s)}$. We note $z^{(s)} = f^{(s)} \circ \cdots \circ f^{(1)}(x)$.

### 3.1 MEMORIZING NORMAL CLASS PROTOTYPES

We first present how memory modules can be used in CL to provide robust and representative normal class prototypes, and then generalize the idea to several scales throughout the encoder.

Foremost, we choose to apply a CL type method rather than other unsupervised learning schemes, as it produces better representations with very few labeled data (Cole et al., 2022). We also favor self-supervised learning only on the normal data rather than using a pre-trained encoder on generic

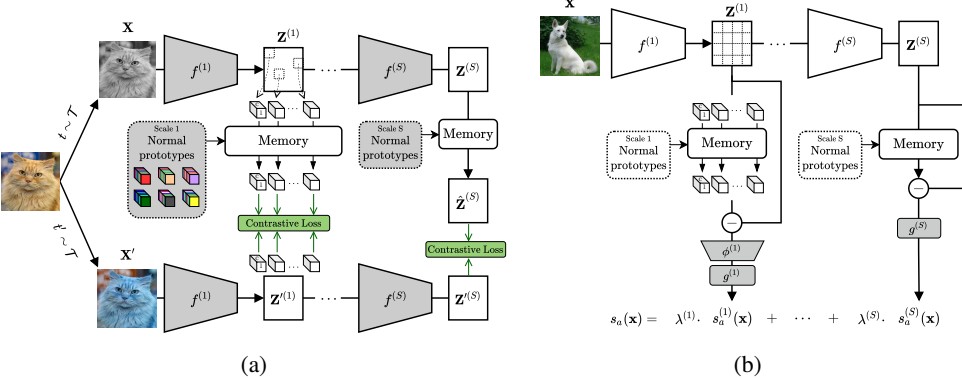

(a)               (b)

Figure 2: Overview of AnoMem's training. (a) First representation learning stage where normal prototypes are learned at multiple scales using Hopfield layers and CL (b) Second anomaly detection stage where multi-scale anomaly distance models are learned from memory deviation maps. Learnable parts of the model are in dark gray.

datasets [2] (Xiao et al., 2021) which often performs poorly on data with a significant distribution shift. In order to learn unsupervised representations and a set of normal prototypes, we could *sequentially* apply CL then perform *k-means clusterisation* and use the cluster centroids as the *normal prototypes*. However, this approach has two main flaws. First, the representation learning step and the construction of prototypes are completely separated. Indeed, it has been shown in several CL methods (Dwibedi et al., 2021; Caron et al., 2020) that the inclusion of a few representative samples in the negative examples greatly improves the representation quality, and alleviate the need for large batches. Moreover, the resulting k-means prototypes do not often cover atypical samples. This means that harder normal samples will not be well encompassed by the normal prototypes, resulting in high false rejection rate during AD. We compare our approach with k-means centroids in Sec. 4.3. To address these flaws, we introduce a novel approach based on memory modules to **jointly learn** an encoder and normal prototypes. Let $z_k$ and $z'_k$ respectively be the encoder features for the contrastive upper and lower branch. Instead of contrasting $z_k$ and $z'_k$, we apply beforehand a Hopfield memory layer $\mathrm{HF}(\cdot)$ to the first branch in the case of normal samples. We note that the memory layer is only applied when the sample is normal, since we assumed that anomalous data is significantly more variable and less defined than normal data. As such, we note

$$\mathrm{Mem}(z, y) = y \cdot \mathrm{HF}(z) + (1 - y) \cdot z \tag{1}$$

With SimCLR loss, we introduce the following contrasted memory loss for representation learning:

$$\mathcal{L}_{\mathrm{COM}}(z, z', y) = \frac{1}{2B} \sum_{k=1}^{B} \left[ \ell_{\mathrm{NTX}} \left( \mathrm{Mem}(z_k, y_k), z'_k \right) + \ell_{\mathrm{NTX}} \left( z'_k, \mathrm{Mem}(z_k, y_k) \right) \right] \tag{2}$$

where $\ell_{\mathrm{NTX}}(z, z') = -\log \frac{\exp(\mathrm{sim}(z,z')/\tau)}{\sum_p \mathbb{1}_{[p \neq z]} \exp(\mathrm{sim}(z,p)/\tau)}$. $p$ covers any representation inside the multi-view batch, $\tau$ is a temperature hyper-parameter and $\mathrm{sim}(\cdot, \cdot)$ is the cosine similarity. In contrast with existing two-stage AD methods, we **explicitly** introduce the anomalous and normal labels from the very first step of representation learning. We note that the labels $y_k$ are only used to exclude anomalies from the memory learning, making the representation learning both usable in OC and OE settings.

**Variance loss as regularization.** Our procedure can be prone to representation collapse during the first epochs. Indeed, we observed that the dynamic between CL and the randomly initialized memory layer can occasionally lead to a collapse of all prototypes to a single point during the first epochs. To prevent this, we introduce an additional regularization loss which ensures the variance of the retrieved memory samples does not reach zero:

$$\mathcal{L}_{\mathrm{V}}(z, y) = -\frac{1}{\sum_k y_k} \sum_{k=1}^{B} y_k \sqrt{\mathrm{Var} \left[ \mathrm{Mem}(z_k, y_k) \right]} \tag{3}$$

**Multi-scale contrasted memory.** To gather information from several scales, we apply our contrasted

---

[2]It is not fair to compare pre-trained methods to ours. AnoMem is trained from scratch with only 50K images (of CIFAR) whereas those methods use an additional large-scale dataset like ImageNet-21K (14M images).

memory loss not only to the flattened 1D output $z$ of our encoder but also to $S$ intermediate layer 3D feature maps $z^{(1)}, \cdots, z^{(S)}$.

We add after each scale representation $z^{(s)}$ a memory layer $\text{HF}^{(s)}$ to effectively capture multi-scale normal prototypes. However, memorizing the full 3D map as a single flattened vector would not be ideal. Indeed, at lower scales we are interested in memorizing local patterns regardless of their position. Moreover, the memory would span across a space of very high dimensions. Therefore, 3D intermediate maps are viewed as a collection of $H^{(s)} W^{(s)}$ 1D feature vectors $z_{i,j}^{(s)}$ rather than a single flattened 1D vector. This is equivalent to remembering the image as patches. Since earlier features map will have a high resolution, the computational cost and memory usage of such approach can quickly explode. Thus, we only apply our contrasted memory loss on a random sample with ratio $r^{(s)}$ of the available vectors on the $s^{\text{th}}$ scale. Our **multi-scale contrasted memory loss** becomes

$$\mathcal{L}_{\text{COM-MS}} = \frac{1}{\sum_s |\Omega^{(s)}|} \sum_{s=1}^{S} \sum_{(i,j) \in \Omega^{(s)}} \lambda^{(s)} \left[ \mathcal{L}_{\text{COM}} \left( z_{i,j}^{(s)}, z'^{(s)}_{i,j}, y \right) + \lambda_V \mathcal{L}_V(z_{i,j}^{(s)}, y) \right] \tag{4}$$

where $\lambda_V$ controls the impact of the variance loss, $\lambda^{(s)}$ controls the importance of the $s^{\text{th}}$ scale, and $\Omega^{(s)}$ is a random sample without replacement of $\lfloor H^{(s)} W^{(s)} r^{(s)} \rfloor$ points from $[\![1, H^{(s)}]\!] \times [\![1, W^{(s)}]\!]$. We choose to put more confidence on the latest stages which are more semantically meaningful than earlier scales, meaning that $\lambda^{(1)} < \cdots < \lambda^{(S)}$.

We simultaneously minimize this loss on all the encoder stages and the $S$ memory layers' weights. An overview of this first stage is given in Fig. 2a, and its algorithm is presented in Alg. 1. Compared to previous memory bank equipped anomaly and OOD detectors (Gong et al., 2019; Park et al., 2020; Bergman et al., 2020; Gui, 2021; Zhang et al., 2023), our model is the first to memorize the normal class at several scales, allowing it to be more robust to anomaly sizes. Moreover, the use of normal memory does not only improve AD but also the quality of the learned representations, as will be discussed in Sec. 4.3.

## 3.2 MULTI-SCALE NORMAL PROTOTYPE DEVIATION

In this second step of training, our goal is to compute an anomaly score given the pre-trained encoder $f$ and the multi-scale normal memory layers $\text{HF}^{(1)}, \cdots, \text{HF}^{(S)}$. For each scale $s$, we consider the difference $\Delta^{(s)}$ between the encoder feature map $z^{(s)}$ and its recollection from the $s^{\text{th}}$ memory layer. The recollection process consists in spatially applying the memory layer to each $C^{(s)}$ depth 1D vector (with $(\text{HF}(z))_{i,j} = \text{HF}(z_{i,j})$):

$$\Delta^{(s)} = z^{(s)} - \text{HF}^{(s)} \left( z^{(s)} \right) \tag{5}$$

**One-class AD.** We use the $L_2$ norm of the difference map as an anomaly score for each scale, and no further training is required:

$$s_a^{(s)}(x) = \|\Delta^{(s)}\|_2 \tag{6}$$

**Outlier-exposure AD.** We use the additional anomaly data to train $S$ scale-specific classifiers on the difference map $\Delta^{(s)}$. Each classifier is first composed of an average pooling layer $\phi$ followed by a two-layer MLP $g^{(s)}$ with a single scalar output. $\phi$ reduces the spatial resolution of $\Delta^{(s)}$, to prevent using very large layers on earlier scales. The output of $g^{(s)}$ directly corresponds to the $s^{\text{th}}$ scale anomaly distance:

$$s_a^{(s)}(x) = g^{(s)} \circ \phi(\Delta^{(s)}) \tag{7}$$

Each scale-specific classifier is trained using the intermediate features of the same normal and anomalous samples used during the first step along their labels. The training procedure is similar to other distance-based anomaly detectors (Ruff et al., 2020; Jezequel et al., 2022a) where the objective is to obtain small distances for normal samples while keeping high distances on anomalies. We note that our model is the *first to introduce memory prototypes learned during representation learning into the anomaly distance learning*. The distance constraint is enforced via a double-hinge distance loss: $\ell_{\text{dist}}(d, y) = y \cdot \max \left( d - \frac{1}{M}, 0 \right) + (1-y) \cdot \max (M - d, 0)$, where $d$ is the anomaly distance for a given sample and $M$ controls the size of the margin around the unit ball frontier. Using this loss, both

normal samples and anomalies will be correctly separated without encouraging anomalous features to be pushed toward infinity. Our **second stage loss** is:

$$\mathcal{L}_{\text{SUP}} = \frac{1}{S \cdot B} \sum_{k=1}^{B} \sum_{s=1}^{S} \ell_{\text{dist}} \left( g^{(s)} \circ \phi(\Delta^{(s)}), y_k \right) \tag{8}$$

Finally, all scale anomaly scores are merged using a sum weighted by the confidence parameters $\lambda^{(s)}$:

$$s_a(\boldsymbol{x}) = \frac{1}{\sum_s \lambda^{(s)}} \sum_s \lambda^{(s)} \cdot s_a^{(s)}(\boldsymbol{x}) \tag{9}$$

The anomaly score $s_a$ effectively combines the expertise of the scores from each scale, making it more robust to different sizes of anomalies than other detectors. As mentioned in Sec. 3.1, the $\lambda^{(s)}$ will put more weight to later scales, making our detector more sensitive to broad object anomalies. This second stage is summarized in Fig. 2b. As we can see, only the second stage has to be swapped between OC and OE learning, resulting in a unified easily-switchable framework for AD.

## 4 EXPERIMENTS

### 4.1 EVALUATION PROTOCOL AND MODEL DESIGN

In the first **one-vs-all** protocol, one class is considered as normal and the others as anomalous. The final reported result is the mean of all runs obtained for each possible normal class. We consider various ratios $\gamma$ of anomaly data in the training dataset and for each, average the metrics on 10 random samples. The OCAD setting is a special case of the OEAD setting with $\gamma = 0$. This protocol presents a *realistic challenge* and is widely used in AD literature. It shows the model ability to *generalize* to different anomalies *without* using a *significant amount* of training anomalies. The datasets considered include CIFAR-10, CIFAR-100 (Krizhevsky, 2009), and CUB-200 (Welinder et al., 2010). It is worth noting that CUB-200 is a particularly challenging, featuring 200 fine-grained bird classes, each with about 50 images. **SOTA AD methods like CSI struggle** to perform effectively on this dataset (see Tab. 1). Our evaluation encompasses not only object anomalies but also *more subtle style and local anomalies* where classes differ on minute details.

The second protocol of intra-dataset cross-type is centered around **face presentation attack detection (FPAD)**, aiming to distinguish between real and fake faces. Training and test data are sampled from the same dataset, albeit with one tested attack type being unseen during training. We can thus evaluate the model generalization power and robustness to unseen attack types. Several attacks are considered: paper print (PP), screen recording (SR), paper mask (PM) and flexible mask (FM). We use the WMCA dataset (George et al., 2020), which contains over 1900 RGB videos of real faces and presentation attacks, spanning various attack types, including object, style, and local anomalies.

For the final protocol of **OOD detection**, we evaluate how well the model discriminate ID samples from OOD samples, after learning on the whole ID dataset. For this evaluation, we add the commonly used datasets **SVHN** (Netzer et al., 2011) and **LSUN** (Yu et al., 2015).

We report the area under the ROC curve (AUROC) averaged over all normal classes in the case of one-vs-all, and the relative error reduction $((1-\alpha_2)-(1-\alpha_1))/(1-\alpha_1)$ to compare two AUROCs $\alpha_1$ and $\alpha_2$.

**Model design.** Regarding network architecture, a Resnet-50 (He et al., 2016) ($\approx 25M$ parameters) is used as the backbone $f$. We consider two different memory scales: one after the third stage and another after the last stage. The associated memory layers are respectively of size $N_{\text{Mem}}^{(1)} = 512$ and $N_{\text{Mem}}^{(2)} = 256$ with an inverse temperature $\beta = 2$, along with a pattern sampling ratio of $r^{(1)} = 0.3$ and $r^{(2)} = 1$. The choices of memory size and sampling ratios are respectively discussed in Sec. 4.4 and Sec. 4.5. The scale confidence factors are set to increase exponentially as $\lambda^{(s)} = 2^{s-1}$, and the variance loss factor is fixed to $\lambda_V = 0.05$ after optimization on CIFAR-10. We use $\tau = 0.1$ as suggested in (Chen et al., 2020). For the anomaly distance loss, we choose a margin size of $M = 2$.

Implementation details and the selection of hyperparameters can be found in Appendix.

Table 1: Comparison with SoTA in one-vs-all protocol. The three blocks respectively contain OC, OE and methods usable in both settings. AnoMem is the best performing unified model. (We re-evaluated Elsa, DP-VAE, GOAD and ARNet on CIFAR100 and CUB-200. * is from (Mirzaei et al., 2023). + is included only for reference, not for a direct performance comparison, as it uses very large external datasets—significantly more than our OE setting where we achieved better results.)

| AUROC (%) | CUB-200 | | | | CIFAR-10 | | | | CIFAR-100 | | | |
|---|---|---|---|---|---|---|---|---|---|---|---|---|
| Models \ $\gamma$ | 0. | 0.01 | 0.05 | 0.10 | 0. | 0.01 | 0.05 | 0.10 | 0. | 0.01 | 0.05 | 0.10 |
| MemAE (Gong et al., 2019) | 59.6 | | | | 60.9 | | | | 57.4 | | | |
| PIAD (Tuluptceva et al., 2019) | 63.5 | | | | 79.9 | | | | 78.8 | | | |
| GOAD (Bergman & Hoshen, 2020) | 66.6 | | | | 88.2 | | | | 74.5 | | | |
| MHRot (Hendrycks et al., 2019) | 77.6 | | | | 89.5 | | | | 83.6 | | | |
| Reverse Distillation (Deng & Li, 2022) | - | | | | 86.5 | | | | - | | | |
| SSD (Sehwag et al., 2021) | - | | | | 90.0 | | | | 85.1 | | | |
| CSI (Tack et al., 2020) | 52.4* | | | | **94.3** | | | | 85.8 | | | |
| *MSC*+ (Reiss & Hoshen, 2023) | - | | | | 94.8+ | | | | 94.4+ | | | |
| Supervised | | 53.1 | 58.6 | 62.4 | | 55.6 | 63.5 | 67.7 | | 53.8 | 58.4 | 62.5 |
| Elsa (Han et al., 2021) | | 77.8 | 81.3 | 82.9 | | 80.0 | 85.7 | 87.1 | | 81.3 | 84.6 | 86.0 |
| DeepSAD (Ruff et al., 2020) | 53.9 | 62.7 | 63.4 | 65.1 | 60.9 | 72.6 | 77.9 | 79.8 | 56.3 | 67.7 | 71.6 | 73.2 |
| DP VAE (Daniel et al., 2019) | 61.7 | 65.4 | 67.2 | 69.6 | 52.7 | 74.5 | 79.1 | 81.1 | 56.7 | 68.5 | 73.4 | 75.8 |
| AnoMem (ours) | **81.4** | **84.1** | **85.3** | **86.0** | 91.5 | **92.5** | **97.1** | **97.6** | **86.1** | **90.9** | **92.3** | **94.7** |

## 4.2 COMPARISON TO THE STATE-OF-THE-ART

### 4.2.1 ONE-VS-ALL

Considered OC methods are reconstruction error generative model (Tuluptceva et al., 2019), the knowledge distillation method (Deng & Li, 2022), pretext tasks methods (Bergman & Hoshen, 2020; Hendrycks et al., 2019), and the two-stage method (Tack et al., 2020). We also consider the two-stage OE-AD (Han et al., 2021). To further show the disadvantages of classical binary classification, we also include a classical deep classifier trained with batch balancing between normal samples and anomalies. For the sake of completeness, we also include the pre-trained model on large dataset MSC Reiss & Hoshen (2023). Lastly, unified methods usable in both OC and OE settings are included with the reconstruction error model (Daniel et al., 2019), and direct anomaly distance models (Ruff et al., 2020). For a fair comparison in the same conditions, we take the existing implementations or re-implement and evaluate ourselves all OC methods, except (Fei et al., 2020; Bergman & Hoshen, 2020; Tack et al., 2020). The results are presented in Tab. 1.

First, the classical supervised approach falls far behind anomaly detectors on all datasets. This highlights the importance of specialized AD as classical models are likely to overfit on anomalies.

Furthermore, AnoMem overall outperforms significantly all considered detectors with up to 62% relative error improvement on CIFAR-10 and $\gamma = 0.01$. Although performance greatly increases with more abnormal data, it remains highly competitive with only normal samples. For OC-AD, AnoMem outperforms all methods specialized for OC including pretext task and reconstruction error methods. The usage of memory in AnoMem is much more efficient than the memory for reconstruction in MemAE. Indeed, while we learn the memory through CL, MemAE and others (Gong et al., 2019; Park et al., 2020) learned it via the pixel-wise reconstruction loss. Their normal prototypes are much more constrained and therefore less semantically rich and generalizable. For OE-AD, AnoMem reduces SoTA error gap on nearly all anomalous data ratio. Its multi-scale anomaly detectors allow capturing more fine-grained anomalies as we can see in the CUB-200 results.

Finally, AnoMem performs very well in both OC- and OE-AD while other unified methods fail in OC-AD. To the best of our knowledge, AnoMem is the *first efficient unified anomaly detector*. We also note that our change from OC-AD to OE-AD was done with minimal hyperparameter tuning. This is due to the first training step being shared between OC and OE settings.

### 4.2.2 OUT-OF-DISTRIBUTION (OOD) DETECTION

As can be seen in Tab. 2, AnoMem performs similarly well as the SoTA baseline CSI (Tack et al., 2020), however the latter mainly relies during inference on keeping in memory the entire training set features. This results in a very high memory footprint on huge training sets, while our model ingeniously learns a fixed size memory[3]. Moreover, CSI often performs poorly on datasets with a few training normal samples as can be seen on one-vs-all CUB-200 (50 images per class) results.

---

[3]In terms of memory usage, CSI and pretrained models using KNN require the storage of the entire normal set, whereas AnoMem only needs a fixed size: for OOD detection with CIFAR as ID, CSI and pretrained models need to store 60000 samples, whereas ours only requires 768 samples (256+512 for two memory modules).

Table 2: Comparison of AnoMem with SoTA methods on OOD detection protocol.

| AUROC (%) | CIFAR-10 (ID) → | | |
|---|---|---|---|
| Models / OOD data | SVHN | LSUN | CIFAR-100 |
| GOAD (Bergman & Hoshen, 2020) | 96.3 | 89.3 | 77.2 |
| MHRot (Hendrycks et al., 2019) | 97.8 | 92.8 | 82.3 |
| FS (Zaeemzadeh et al., 2021) | 95.3 | 96.2 | 84.8 |
| Energy OE (Liu et al., 2020) | 96.6 | 94.0 | 84.4 |
| Energy PEBAL (Choi et al., 2023) | 97.7 | 94.5 | 85.2 |
| CSI (Tack et al., 2020) | 99.8 | 97.5 | 89.2 |
| AnoMem (ours) | 99.2 | 97.5 | 89.2 |

Table 3: Comparison of AnoMem with SoTA in FPAD, columns represent unseen attacks.

| AUROC (%) | WMCA | | | | |
|---|---|---|---|---|---|
| Models / Kind | All | PP | SR | PM | FM |
| PIAD (Tuluptceva et al., 2019) | 76.4 | | | | |
| MHRot (Hendrycks et al., 2019) | 81.3 | | | | |
| CSI (Tack et al., 2020) | 82.7 | | | | |
| MSC (Reiss & Hoshen, 2023) | 85.3 | | | | |
| GOAD (Bergman & Hoshen, 2020) | 86.1 | | | | |
| Supervised | | 78.3 | 77.1 | 80.7 | 81.9 |
| Elsa (Han et al., 2021) | | 86.1 | 84.3 | 89.2 | 89.1 |
| DP VAE (Daniel et al., 2019) | 53.9 | - | - | - | - |
| DeepSAD (Ruff et al., 2020) | 71.2 | 79.9 | 80.3 | 81.8 | 83.4 |
| AnoMem (ours) | 86.9 | 91.3 | 89.8 | 93.0 | 92.7 |

Table 4: Performance of AnoMem for image classification through linear evaluation.

| Memory | Multi-scale | CIFAR-10 | CIFAR-100 |
|---|---|---|---|
| - | - | 88.2 | 80.5 |
| ✓ | - | 91.4 (+3.2) | 84.1 (+3.6) |
| ✓ | ✓ | 91.9 (+0.5) | 84.8 (+0.7) |

Table 5: Performance of AnoMem for OC-AD with multi-scale memory.

| Memory | Multi-scale | CIFAR-10 | CIFAR-100 | WMCA |
|---|---|---|---|---|
| - | - | 88.1 | 82.7 | 81.6 |
| - | ✓ | 89.4 (+1.3) | 83.8 (+1.1) | 84.0 (+2.4) |
| ✓ | - | 90.5 (+2.4) | 85.3 (+2.6) | 84.8 (+3.2) |
| ✓ | ✓ | 91.5 (+1.0) | 86.1 (+0.8) | 86.9 (+2.1) |

It is also worth noting that algorithms in (Reiss & Hoshen, 2023; Li et al., 2023) obtain higher performance on this task but they rely on models pretrained on an additional large-scale dataset such as ImageNet-21K (**14M** images) (Russakovsky et al., 2015). Therefore, comparing directly those methods to AnoMem, being trained from scratch on a smaller dataset (**50K** images of CIFAR), **is not fair** as there is an overlap in nature of the pre-training samples used in (Li et al., 2023) and anomalous samples to be detected. Finally, we note that in this protocol the normal class is composed of several subclasses. This shows the ability of our memory modules to cover multi-modal normal cues.

### 4.2.3 FACE PRESENTATION ATTACK DETECTION (FPAD)

Tab. 3 compares our model on the FPAD intra-dataset cross-type protocol [4] with methods presented in Sec. 4.2.1 (we re-evaluated those methods for FPAD using official codes). Without any further tuning for face data, our method improves FPAD performance on WMCA with an error relative improvements of up to 14% on paper prints. It outperforms existing anomaly detectors on all unseen attack type, including the OC setting. We can also notice that it reduces the error gap between coarse attacks (PM, FM) and harder fine-grained attacks (PP, SR) thanks to its multi-scale AD. AnoMem outperforms CSI and pre-trained method MSC. Moreover, the utilization of certain large-scale datasets, e.g., ImageNet-21K, may not be permitted in industrial applications due to restrictive licenses.

### 4.3 ABLATION STUDY

This section studies the impact of the multi-scale memory layers in the two training stages and show they are essential to our model performance. The baseline is the model using only the last feature map, with k-means centroids as its normal prototypes.

First, we assess the impact of memory through linear evaluation on standard CIFAR-10/-100 classification benchmarks, examining how it influences the CL of the encoder representations. This involves training an extra linear classifier on the frozen representations of the encoder. It is worth noting that this experiment focuses on multi-class image classification rather than AD. The goal is to validate the efficacy of the first step in our overall approach shown in Fig. 2a. As shown in Tab. 4, the inclusion of the memory layers on the first branch drastically improves the quality of the encoder representations. We hypothesize that, as shown in (Dwibedi et al., 2021), the inclusion of prototypical samples in one of the branch allows to contrast positive images against representative negatives. This alleviates the need for large batch size, and highly reduces the multi-scale CL memory usage.

To support the importance of memory during AD, we compare the anomaly detector with k-means centroids or with the normal prototypes learned during the first stage (Tab. 5). In the first case, we train the anomaly detectors with the same procedure but instead of fetching the Hopfield layer output we use the closest k-means centroid. Lastly, we measure the impact of multi-scale AD by comparing the single-scale model only using the last feature map, and our two-scale model. While we sacrifice

---

[4]A few works have sought to apply **generic** AD to this real-world challenging problem with local anomalies.

some of the memory and training time for the additional scale, our AD performance is improved significantly with up to 10% error relative improvement on CIFAR-10.

## 4.4 MEMORY SIZE

The memory size must be carefully chosen at every scale to reach a good balance between the normal class prototype coverage and the memory usage during training and inference. This section presents some rules of thumb regarding the sizing of memory layers depending on the scale.

We start by plotting the relation between the last scale memory size, the representation separability, and the final AD performance in Fig. 3a. As one could expect, higher memory size produces better quality representations and more accurate anomaly detector. However, we can note that increasing the memory size above 256 has significantly less impact on the anomaly detector performance. Therefore a good trade-off between memory usage and performance seems to be at 256 on CIFAR-10.

Further, we study the impact of feature map scale on the required memory size. In Fig. 3b we fix the last scale memory size to 256 and look at the AD accuracy for various ratios of memory size between each scale. As lower scales benefit more from a larger memory than higher scales, we hypothesize that the feature vectors of more local texture-oriented features are richer and more variable than global object-oriented ones. Memory layers thus need to be of higher capacity to capture the complexity. We further show the effectiveness of our memory mechanism in the appendix.

## 4.5 SPATIAL SAMPLING RATIOS

Sampling ratios $r$ are introduced in the first step to reduce the amount of patterns used in the contrastive loss, thereby reducing the size of the similarity matrix. At lower scales, similar nearby samples make it less detrimental to skip some available patterns during training.

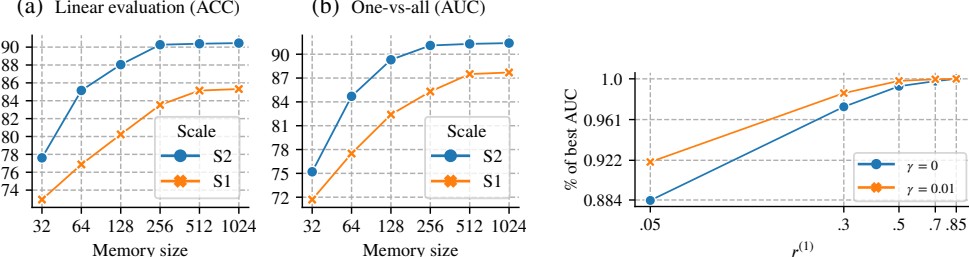

Figure 3: Memory experiments on CIFAR10.     Figure 4: Sampling ratio experiments.

To guide our choice of sampling ratio, we measure our anomaly detection AUC with various sampling ratios and anomalous data ratio $\gamma$. Since the last scale feature maps are spatially very small, we only vary the first scale ratio $r^{(1)}$ and set $r^{(2)} = 1$. The batch size if fixed throughout the experiments. The results are displayed in Sec. 4.5. We can see that low sampling ratios ($\gamma < 0.3$) significantly decrease the AD performance. However, the gain in performance for higher ratios is generally not worth the additional computational cost: by more than doubling the amount of sampled patterns, we only increase the relative AUC by 2%.

## 5 CONCLUSION AND FUTURE WORK

In this paper, we introduce a novel two-stage AD model, AnoMem, designed to seamlessly operate in both OC and OE settings. Our approach involves the memorization of prototypes for normal class instances to compute an anomaly deviation score. By incorporating learnable memory layers for normal instances within a CL framework, we jointly enhance encoder representations and establish robust memorization of normal samples. Subsequently, these learned normal prototypes are leveraged to train a straightforward detector within a unified framework that accommodates both OC-AD and OE-AD scenarios. Moreover, we extend these prototypes to multiple scales, enhancing the model's robustness against various anomaly sizes. Finally, we evaluate the performance of AnoMem on diverse datasets containing object, style, and local anomalies to demonstrate its efficiency, even when trained from scratch under limited data regimes.

For future work, we could explore the use of our model for anomaly localization. Indeed, we could investigate how to integrate more efficiently additional scales and merge the anomaly maps into a single heatmap. One solution could be to use Barlow-Twins instead of SimCLR by replacing $\ell_{NTX}$ with the cross-correlation.

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

APPENDIX

## A FIRST TRAINING STAGE OF ANOMEM (REPRESENTATION LEARNING)

---
**Algorithm 1** AnoMem first learning stage
---
1: **Input:** batch size $B$, invariance transformations $\mathcal{T}$
2: **Initialization:** encoders $f^{(1)}, \cdots, f^{(S)}$, memory $\mathrm{HF}^{(1)}, \cdots, \mathrm{HF}^{(S)}$.
3: **while** not reach the maximum epoch **do**
4:      Sample image minibatch $\mathbf{X}$ with labels $\mathbf{y}$
5:      Sample augmentations $t, t'$ from $\mathcal{T}$
6:      Get augmented views $\mathbf{Z}^{(0)} \leftarrow t(\mathbf{X})$ and $\mathbf{Z}'^{(0)} \leftarrow t'(\mathbf{X})$
7:      **for** $s = 1 \cdots S$ **do**
8:          $\mathbf{Z}^{(s)} \leftarrow f^{(s)}(\mathbf{Z}^{(s-1)})$ and $\mathbf{Z}'^{(s)} \leftarrow f^{(s)}(\mathbf{Z}'^{(s-1)})$
9:          Sample $\lfloor H^{(s)} W^{(s)} r^{(s)} \rfloor$ vectors $Z_{i,j}^{(s)}$ from $\mathbf{Z}^{(s)}$
10:         Retrieve each $Z_{i,j}^{(s)}$ memory prototypes using Eq. (1) with the $s^{\text{th}}$ scale memory layer.
11:      Compute $\mathcal{L}_{\text{COM-MS}}$ from Eq. (4)
12:      Gradient descent on $\mathcal{L}_{\text{COM-MS}}$ to update $f^{(1)}, \cdots, f^{(S)}$ and $\mathrm{HF}^{(1)}, \cdots, \mathrm{HF}^{(S)}$.
13: **Output: Encoder network** $f$, and the **multi-scale memory prototypes** from $\mathrm{Mem}^{(1)}, \cdots, \mathrm{Mem}^{(S)}$.
---

## B IMPLEMENTATION DETAILS

**Optimization.** Training is performed under SGD optimizer with Nesterov momentum (Sutskever et al., 2013), using a batch size of $B = 1024$ and a cosine annealing learning rate scheduler (Loshchilov & Hutter, 2017) for both of the stages.

**Data augmentation.** For the contrastive invariance transformations, we use random crop with rescale, horizontal symmetry, brightness jittering, contrast jittering, saturation jittering with Gaussian blur and noise as in SimCLR (Chen et al., 2020). It is worth noting that we do not need specific augmentations as in CSI (Tack et al., 2020). Indeed, CSI requires *additional* shift transformations (alongside standard augmentations like SimCLR and AnoMem) to generate its pseudo OOD negatives. AnoMem performs *differently*: it directly learns prototypes of normal samples and does *not* rely on pseudo negatives or require shift augmentations.

**Model design.** We conducted a performance evaluation of AnoMem using different backbone architectures, including EffNet-B0, ResNet18, and ResNet50. In the context of one-vs-all evaluation settings (OC and OE) on the CIFAR dataset, we achieved consistently high performance.

EffNet-B0 emerged as the top performer, demonstrating superior results, followed closely by ResNet50 and then ResNet18. The performance gaps between these architectures were relatively small, with EffNet-B0 outperforming ResNet50 by a margin of 0.2% to 0.3%, while ResNet50 surpassed ResNet18 by a modest margin of 0.4% to 0.6%. In our main paper, we presented the performance of AnoMem using ResNet50, which was the median performer. It is worth noting that the memory modules are scalable with the chosen backbone.

## C SELECTION AND TUNING OF HYPERPARAMETERS

The variance loss weight $\lambda_V$ was evaluated on CIFAR10 using different commonly used values. As shown in Tab. 6, AnoMem is not sensitive to $\lambda_V$. Moreover, in our tests, it remains *fixed* for all other datasets.

For the confidence weights $\lambda^{(s)}$, we compared two monotonically increasing functions linear and exponential. As we can see from the results in Tab. 7, the exponential weighting yields a better anomaly detection than the linear weights. We believe that the linear relation does not sufficiently enhance object anomalies and imposes excessive constraints during representation learning, particularly when contrasting very low-scale patterns.

The margin $M$ enforces a norm constraint on normal and anomaly representations within a radius of less than $1/M$ and more than $M$, respectively. A value of 1 pushes all representations indiscrimi-

nately onto the unit sphere, while a high value leads to an ill-posed distance loss. We chose $M = 2$ as a compromise, allowing half of the unit sphere to be allocated for normal samples. Indeed, we are currently working on dynamically adjusting the value of $M$.

Table 6: Linear evaluation on CIFAR-10 for different $\lambda_V$.

| $\lambda_V$ | 0.1 | 0.05 | 0.025 | 0.01 |
|---|---|---|---|---|
| AUROC (%) | 91.90 | 91.91 | 91.87 | 91.86 |

Table 7: OC-AD evaluation on CIFAR-10 and CIFAR-100 for different $\lambda^{(s)}$ functions.

| $\lambda^{(s)}$ | CIFAR-10 | CIFAR-100 | avg. |
|---|---|---|---|
| **linear** | 91.2 | 85.6 | 88.4 |
| **exp** | 91.5 | 86.1 | 88.8 |

## D QUALITATIVE EFFECTIVENESS OF LEARNED PROTOTYPE

We perform in Fig. 5 a t-SNE analysis on the last-scale prototypes, along with test samples from normal and anomalous classes of CIFAR-10 after the first learning stage of AnoMem. As we can see, the representations of normal and anomalous samples are well separated, confirming the effectiveness of our memory backed representation learning. Moreover, the learnt normal prototypes are well representative of the normal class, as testified by the close overlap in representation space.

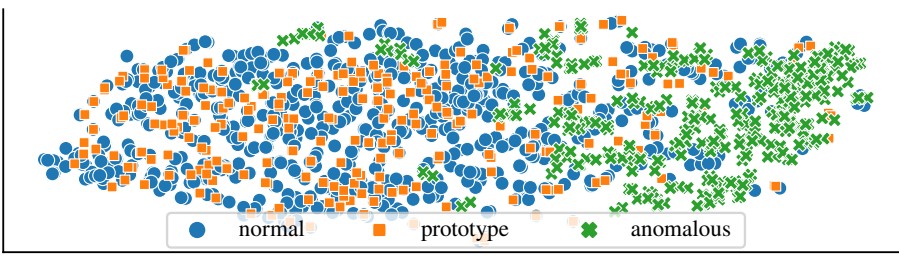

Figure 5: t-SNE of learnt prototypes, normal and anomalous samples on CIFAR10.

