# OpenReview forum: "Unified Anomaly Detection via Multi-Scale Contrasted Memory"
_ICLR.cc/2024/Conference — Submitted to ICLR 2024_

### Official Review · Reviewer_t6Z1 · 2023-10-15

**Soundness:** 2 fair
**Presentation:** 3 good
**Contribution:** 2 fair
**Rating:** 5
**Confidence:** 5

**Summary:**

The paper presents a solution to address the challenges of one-class anomaly detection and the outlier-exposure scenario, where labeled anomalies are scarce in the training dataset. The authors introduce a method that incorporates a memory module, employing Hopfield layers, which is integrated with contrastive learning techniques. The approach allows for the memorization of multi-scale normal class prototypes during training, but also facilitates the learning of informative representations. This innovation significantly enhances the model's ability to capture subtle features of normal data while adapting to varying levels of anomaly complexity.

**Strengths:**

The authors provide a well-structured and clear motivation for their proposed method. Moreover, the introduction of Hopfield layers for anomaly detection represents a novel concept. This innovative utilization adds efficiency to the model's memory capabilities.

**Weaknesses:**

The paper demonstrates several areas where it could be improved.

1. While the authors assert that their model outperforms state-of-the-art approaches across a wide range of anomalies, it is notable that the paper lacks evaluation on widely-accepted benchmark datasets such as MVTec [1] and VisA [2]. These datasets contain texture anomalies, which are crucial for a comprehensive evaluation. Although CIFAR-10/100 and CUB are important, their evaluation should be complemented by assessments on these texture-oriented datasets.

2. The paper mentions superior performance over existing methods, but Table 2 reveals that AnoMem, at best, achieves comparable results in out-of-distribution (OOD) detection. It's important to ensure that the claims made align with the empirical findings.

3. While it may be justifiable not to compare with state-of-the-art pretrained approaches like [3], given their exposure to external datasets, it remains important to include these comparisons in the evaluation. CIFAR and CUB datasets share similarities with ImageNet-pretrained data, but the FPAD dataset exhibits a significant distribution shift, which calls for a comparative analysis. Additionally, it's worth noting that AnoMem's performance benefits significantly from exposure to anomaly types, which pretrained methods do not rely on.

4. Several crucial components of the proposed method are not examined by the authors. For example, while the authors briefly mention the potential substitution of the NTX objective with alternative contrastive frameworks like Barlow-Twins, which are known for their efficiency and ability to operate effectively with smaller batch sizes, this proposition lacks empirical demonstration. The absence of a comparative analysis or experimental results to support this claim leaves a gap in the evaluation of the method's adaptability and robustness across different contrastive learning frameworks.

[1] MVTec AD — A Comprehensive Real-World Dataset for Unsupervised Anomaly Detection, Bergmann et. al, CVPR 2019.

[2] SPot-the-Difference Self-Supervised Pre-training for Anomaly Detection and Segmentation, Zou et. al, ECCV 2022.

[3] Mean-shifted contrastive loss for anomaly detection, Reiss & Hoshen, AAAI 2023.

**Questions:**

1. In FPAD, the CSI method is not included in performance metrics. What are the CSI results?

2. The explanation of the linear evaluation protocol, while briefly described, lacks depth. It would be valuable to have a more detailed elaboration on the rationale behind employing this protocol and why it is considered an important metric in the context of anomaly detection. Moreover, given that this protocol may not be standard within the AD community, it raises curiosity about how other existing methods perform under this evaluation criterion. Could the authors shed light on the performance of competing methods using this protocol for comparison and context? This information would further enrich the assessment of the proposed approach.

---

> ### Author Response · Authors · 2023-11-22
> **Reply to Reviewer t6Z1 (1/2)**
>
> Dear reviewer,
> Thank you for providing constructive reviews and your recognition. We have addressed each of your questions individually in our response and hope these clarifications contribute to an improved rating. We have revised the paper according to your feedback, marking the changes in magenta. Additionally, we've highlighted in brown the text that was part of the initial submission, which may have been overlooked.
>
> We are eager to continue the discussion. Have all your concerns been addressed in our rebuttal, or are there any remaining comments you would like us to consider?
>
> **Question. While the authors assert that their model outperforms state-of-the-art approaches across a wide range of anomalies, it is notable that the paper lacks evaluation on widely-accepted benchmark datasets such as MVTec and VisA. These datasets contain texture anomalies, which are crucial for a comprehensive evaluation. Although CIFAR-10/100 and CUB are important, their evaluation should be complemented by assessments on these texture-oriented datasets.**
>
> **Reply:**
> Our extensive tests cover a wide range of anomalies, including coarse object anomalies (CIFAR), subtle style anomalies (CUB200, a challenging fine-grained dataset for anomaly detectors with random performance of Sota method like CSI [1]; while recent method MSC [2] does not report their performance on this dataset), and a real-world dataset of face presentation attack WMCA with local anomalies. We would like to highlight that the WMCA conveys different texture anomalies with different forms of attacks, i.e., paper print (PP), screen recording (SR), paper mask (PM) and flexible mask (PM).
>
> With respect to datasets such as MVTec and VisA, only a few studies have employed them for evaluating anomaly detection. Notably, these datasets are primarily designed for anomaly localization purposes. We intend to consider the localization task in future work, as mentioned at the end of the paper.
>
> [1] Tack et al., “CSI: Novelty detection via contrastive learning on distributionally shifted instances”, NeurIPS 2020.
>
> [2] Tal Reiss and Yedid Hoshen, “Mean-shifted contrastive loss for anomaly detection”. AAAI 2023
>
> **Question. The paper mentions superior performance over existing methods, but Table 2 reveals that AnoMem, at best, achieves comparable results in out-of-distribution (OOD) detection. It's important to ensure that the claims made align with the empirical findings.**
>
> **Reply:**
> Thank you for your effort for the detailed review. Indeed, we were already aware of the positioning of our results in the literature for OOD detection.
>
> We have thoroughly reviewed the initial submission and found that we did not explicitly claim superior performance over existing methods for OOD detection. For example, in the abstract, we stated, 'Our model outperforms the state-of-the-art on a wide range of anomalies, including object, style, and local anomalies, as well as face presentation attacks' without specifically mentioning superior performance for OOD. In Section 4.2.2, we began by stating, 'AnoMem performs similarly well as the state-of-the-art baseline CSI.' To eliminate any possible confusion, we have now explicitly mentioned the performance of our method for OOD in the abstract.
> It is worth noting that for OOD detection with CIFAR10 as ID, CSI and pretrained models need to store 60000 samples, whereas ours only requires 768 samples.
>
>
> **Question. While it may be justifiable not to compare with state-of-the-art pretrained approaches like MSC, given their exposure to external datasets, it remains important to include these comparisons in the evaluation. CIFAR and CUB datasets share similarities with ImageNet-pretrained data, but the FPAD dataset exhibits a significant distribution shift, which calls for a comparative analysis. Additionally, it's worth noting that AnoMem's performance benefits significantly from exposure to anomaly types, which pretrained methods do not rely on.**
>
> **Reply:**
> Thank you for your suggestion.
> We could not find the results of MSC [3] for OOD detection with CIFAR10 as ID, and therefore, we are unable to include them. It is worth highlighting that, although classified as an OC method in Table 1, MSC used ImageNet-1K, signifying its use of approximately 1000 times more data than our OE setting, where we achieved better results.
>
> We have now included the results of pretrained method MSC [3] for FPAD on WMCA in Table 3. As hypothesized in your remark, the performances of the pretrained method on ImageNet are impacted by the distribution shift.
> Furthermore, we note that AnoMem only benefits from anomalies in the outlier exposure setting, but not in the one-class setting where we still obtain SOTA performance when compared to pre-trained method.

---

> > ### Author Response · Authors · 2023-11-22
> > **Reply to Reviewer t6Z1 (2/2)**
> >
> > **Question. Several crucial components of the proposed method are not examined by the authors. For example, while the authors briefly mention the potential substitution of the NTX objective with alternative contrastive frameworks like Barlow-Twins, which are known for their efficiency and ability to operate effectively with smaller batch sizes, this proposition lacks empirical demonstration. The absence of a comparative analysis or experimental results to support this claim leaves a gap in the evaluation of the method's adaptability and robustness across different contrastive learning frameworks.**
> >
> > **Reply:**
> > We do recognize that our initial statement might have been misinterpreted as an affirmation of our framework working with all contrastive methods. We only wanted to mention the fact that our contrastive memory learning can easily be integrated into a two-branched contrastive approach such as Barlow twins, BYOL or SimSiam.
> >
> > Due to the limited time of the rebuttal phase, we removed our potentially misleading statement from the method presentation, and reframed it in future work as a direction to improve the efficiency of multiscale contrastive learning.
> >
> > **Question. In FPAD, the CSI method is not included in performance metrics. What are the CSI results?**
> >
> > **Reply:**
> > We have now incorporated the results of CSI for FPAD using its official code in Table 3. Once again, AnoMem demonstrates an important performance advantage over CS in this challenging task. It is also worth noting that in CUB, AnoMem significantly outperforms CSI, showing improvements of nearly 30 points.
> >
> > **Question. The explanation of the linear evaluation protocol, while briefly described, lacks depth. It would be valuable to have a more detailed elaboration on the rationale behind employing this protocol and why it is considered an important metric in the context of anomaly detection. Moreover, given that this protocol may not be standard within the AD community, it raises curiosity about how other existing methods perform under this evaluation criterion. Could the authors shed light on the performance of competing methods using this protocol for comparison and context? This information would further enrich the assessment of the proposed approach.**
> >
> > **Reply:**
> > Thank you for your question.
> >
> > In fact, the linear evaluation experiment focuses on multi-class image classification rather than anomaly detection. The objective is to validate the efficacy of the first step in our overall approach. In more detail, once the representation 'z' is learned, we freeze it and train an additional linear classifier. We then evaluate the entire model (frozen encoder + linear classifier) for image classification tasks using CIFAR-10 and CIFAR-100. The initial submission may have contained some ambiguities due to the merging of results from two distinct tasks. To address this, we have now separated the results for different tasks into two distinct Tables 4 and 5.

---

> > > ### Comment · Reviewer_t6Z1 · 2023-11-23
> > >
> > > Thanks for providing detailed results and for responding to my concerns. I have read the author's responses and all other reviews, but I am keeping my rating the same.

---

### Official Review · Reviewer_xRL9 · 2023-10-28

**Soundness:** 3 good
**Presentation:** 3 good
**Contribution:** 2 fair
**Rating:** 6
**Confidence:** 3

**Summary:**

The paper analyzes the two most common problems in anomaly detection - the one-class problem and the out-of-distribution problem. It proposes a multi-scale contrastive learning framework for anomaly detection to address these two problems. Regarding the one-class problem and the out-of-distribution problem, the authors believe that the main difference between the two is the scale of the anomaly, and using memory to store normal features at different scales can solve both problems simultaneously. To enhance the detection ability of the model, the paper also introduces a contrastive learning framework to train a feature extractor from scratch to obtain class-sensitive features.

**Strengths:**

+ The paper summarizes the differences between the one-class problem and the out-of-distribution problem, and proposes to combine features at different scales for anomaly detection accordingly.
+ The paper introduces a contrastive learning framework to enhance the model's perception ability for anomalies at different scales.
+ The paper validates its effectiveness on a one-class classification dataset for the one-class problem and a fake detection dataset for the out-of-distribution problem.

**Weaknesses:**

+ The motivation behind the paper is very direct, but the authors do not further discuss the benefits of unifying the one-class problem and out-of-distribution problem.
+ Although the framework of the paper for the one-class problem and out-of-distribution problem is consistent, the paper adds a classifier for the out-of-distribution problem. This may be due to the different evaluation metrics of the two problems, but it makes the two problems slightly disconnected.

**Questions:**

+ Essentially, the authors weight different scales of features to unify the one-class problem and out-of-distribution problem. However, the parameter selection for weighting seems somewhat arbitrary.
+ The authors mention at the end of the paper that their method can be used for anomaly localization, and the model design does seem to support this. It is possible that the authors did not consider further experiments on this due to unsatisfactory experimental results.

---

> ### Author Response · Authors · 2023-11-22
> **Reply to Reviewer xRL9 (1/1)**
>
> Dear reviewer,
> Thank you for providing constructive reviews. We have addressed each of your questions individually in our response and hope these clarifications contribute to an improved rating. We have revised the paper according to your feedback, marking the changes in magenta. Additionally, we've highlighted in brown the text that was part of the initial submission, which may have been overlooked.
>
> We are eager to continue the discussion. Have all your concerns been addressed in our rebuttal, or are there any remaining comments you would like us to consider?
>
>
> **Question. The motivation behind the paper is very direct, but the authors do not further discuss the benefits of unifying the one-class problem and out-of-distribution problem.**
>
> **Reply:**
> Thank you for the question. We made this point clearer in the revised paper.
> We would like to emphasize that AnoMem stands out as the first to be both unified and generic.
> 1.  In terms of its unified property, specifically regarding the availability of anomaly samples during training, AnoMem excels in both one-class and imbalanced outlier-exposure settings—a crucial practical feature in real-world scenarios where normal samples are initially available, and anomaly samples are acquired later, a scenario not adequately addressed by existing literature.
> 2. Moving on to its generic property, AnoMem demonstrates effectiveness in object-/semantic-level anomaly detection, out-of-distribution (OOD) detection, and also in industrial applications like fine-grained anomaly detection (FPAD).
> 3. While not the primary focus of our paper, we further demonstrate that our learning method can enhance the more general task of image classification. This aspect might have been overlooked in the initial submission, where different results for distinct tasks were presented in the same Table 4. To provide a clearer depiction of its capabilities, we have now separated the results for different tasks into Tables 4 and 5.
>
>
> **Question. Although the framework of the paper for the one-class problem and out-of-distribution problem is consistent, the paper adds a classifier for the out-of-distribution problem. This may be due to the different evaluation metrics of the two problems, but it makes the two problems slightly disconnected.**
>
> **Reply:**
> Thank you for your question.
> We did NOT include a classifier specifically for out-of-distribution detection. The additional linear classifier is solely trained to evaluate our learned representation z on image classification tasks using CIFAR-10 and CIFAR-100. The initial submission may have contained some ambiguities due to the merging of results from two distinct tasks, as mentioned earlier. To address this, we have now separated the results for different tasks into two distinct tables (please refer to Tables 4 and 5).
>
> **Question. Essentially, the authors weight different scales of features to unify the one-class problem and out-of-distribution problem. However, the parameter selection for weighting seems somewhat arbitrary.**
>
> **Reply:**
> For the choice of weights for each scale, we considered two types of monotonically increasing functions to give more weight to the latter scales in which we have more confidence. The results for the two weight functions linear and exponential have been included in Table 7 in Appendix (due to the space limit). As we can see, we obtained better results with the exponential function justifying our choice.
>
> **Question. The authors mention at the end of the paper that their method can be used for anomaly localization, and the model design does seem to support this. It is possible that the authors did not consider further experiments on this due to unsatisfactory experimental results.**
>
> **Reply:**
> Thank you for your remark.
> We do believe that our model could be successfully used for anomaly localization, however not in its current form. Indeed, using the two scales in the latter layers as in our proposed model yields rather low resolution anomaly maps and would have a low AUPRO segmentation metric on datasets such as MVTec. However, as discussed in the paper, adding more scales in earlier layers would result in a substantially slower and larger model. Therefore, the main issue to tackle for an efficient anomaly localization in future work would be to optimize the memory mechanism or use an alternative contrastive scheme that takes less memory per memorized pattern.

---

> > ### Comment · Reviewer_xRL9 · 2023-11-23
> >
> > The author has addressed my concerns, and I am willing to increase my score.

---

### Official Review · Reviewer_9JQe · 2023-11-01

**Soundness:** 2 fair
**Presentation:** 2 fair
**Contribution:** 2 fair
**Rating:** 5
**Confidence:** 4

**Summary:**

This work introduces an approach for contrastive learning of multi-scale memory units for anomaly detection. It applies the idea of contrastive representation learning to prototype-based feature representations for learning multiple memory layers at various intermediate feature layers, and the resulting feature representations can then be used to learn either unsupervised one-class detection models or semi-supervised detection models with some anomaly examples. The approach is evaluated on three one-vs-all one-class classification datasets, one face representation attack detection dataset, and one OOD detection setting using CIFAR-10 as the ID dataset.

**Strengths:**

- The idea of unifying unsupervised one-class anomaly detection and semi-supervised anomaly detection approaches into one framework is interesting. Most methods are focused on the unsupervised case, while some recent studies attempt to tackle a semi-supervised case. I'm not aware of an approach that works well under both cases.
- I appreciate the efforts of bringing ideas from several research lines (contrastive learning, memory learning, OOD detection, and anomaly detection) together to create an effective anomaly detection method.
- The method is evaluated using three different tasks and shows effective performance.
- The method demonstrates good performance on face representation attack detection datasets.

**Weaknesses:**

- It is unclear how much difference it makes by bringing multi-scale learning into the memory learning-based anomaly detection approaches. No appropriate ablation study or empirical comparison is presented there. The baseline in table 4 may be changed to a memory learning method that involves multiple normal prototypes to serve this purpose.
- As demonstrated in table 4, the multi-scale learning has very limited contribution to the overall detection performance.
- It mentions at page 4 about the issue of overfitting in existing methods that use a pre-trained encoder on large-scale image datasets, but this issue should be easily fixed by tuning on the target normal data. I cannot find convincing reasons for not using such well pre-trained encoders, and not comparing with such methods. Additionally, in this work, there is the extensive fitting of the multi-scale memory units to the limited normal data in the target data, which could easily lead to overfitting to the normal data, and so the proposed method could perform poorly on datasets with distribution shift. I wonder whether the authors could perform experiments on datasets with distribution shift to justify their argument.
- The clarity is bad in several aspects. **1)** The memory learning in eqs. 2-4 involves ground truth $y_k$ that could be either $0$ for anomaly or $1$ for normal sample, but in sec. 3.2 there is a one-class AD objective, i.e., eq. 6, where no training anomaly examples are supposed to be available, so I'm confused that how the one-class AD model can be trained together with eqs. 2-4. **2)** The concept of outlier exposure (OE) is defined and used in OOD detection for using external datasets as pseudo OOD examples to train OOD detection models, but in this work it seems it treats real anomaly examples as OE examples. This is very confusing. Rather than using the so-called AD-OE concept, it may be clearer to use semi-supervised AD or open-set AD as in *"Deep semi-supervised anomaly detection. arXiv preprint arXiv:1906.02694."*, *"Catching both gray and black swans: Open-set supervised anomaly detection. In Proceedings of the IEEE/CVF Conference on Computer Vision and Pattern Recognition (pp. 7388-7398)."*, or *"Ubnormal: New benchmark for supervised open-set video anomaly detection. In Proceedings of the IEEE/CVF Conference on Computer Vision and Pattern Recognition (pp. 20143-20153)."* **3)** It is also unclear why we need a method that works well for both anomaly detection and OOD detection, given the fact that the two tasks have quite different application settings.
- Following up the above point, since anomaly examples are used in the training stage, recent SOTA semi-supervised/open-set anomaly detection methods should be used in the experiment comparison to justify the advantages the work has, e.g., see *"Catching both gray and black swans: Open-set supervised anomaly detection. In Proceedings of the IEEE/CVF Conference on Computer Vision and Pattern Recognition (pp. 7388-7398)."* for some of such methods.
- Experiments on large-scale high-resolution image datasets, e.g., the popular setting that uses ImageNet-1k as the ID dataset, are missing for the OOD detection task.

**Questions:**

Pls see the above Weaknesses section for detail.

---

> ### Author Response · Authors · 2023-11-22
> **Reply to Reviewer 9JQe (1/3)**
>
> **Question. It is unclear how much difference it makes by bringing multi-scale learning into the memory learning-based anomaly detection approaches. No appropriate ablation study or empirical comparison is presented there. The baseline in table 4 may be changed to a memory learning method that involves multiple normal prototypes to serve this purpose.**
>
> **Reply:**
> We include in Tab.5 an additional ablation study on the multi-scale component alone to better measure its impact on anomaly detection. As we can see, the multiple scales by themself improve notably the AD performances on CIFAR-10, CIFAR-100 and WMCA.
>
> **Question. As demonstrated in table 4, the multi-scale learning has very limited contribution to the overall detection performance.**
>
> **Reply:**
> The limited impact on the overall detection performances of CIFAR-10 and CIFAR-100 can be explained by the low amount of local anomalies in these datasets. By including representations from earlier layers in the network we can help our model to better encompass smaller and finer anomalies that are more represented in WMCA with partial attacks, as can be seen in the **additional ablation results** on WMCA in Table 5.
>
> **Question. It mentions at page 4 about the issue of overfitting in existing methods that use a pre-trained encoder on large-scale image datasets, but this issue should be easily fixed by tuning on the target normal data. I cannot find convincing reasons for not using such well pre-trained encoders, and not comparing with such methods. Additionally, in this work, there is the extensive fitting of the multi-scale memory units to the limited normal data in the target data, which could easily lead to overfitting to the normal data, and so the proposed method could perform poorly on datasets with distribution shift. I wonder whether the authors could perform experiments on datasets with distribution shift to justify their argument.**
>
> **Reply:**
> In Table 3, we have included additional results for the state-of-the-art pretrained method MSC [Reiss AAAI’23] on WMCA. As can be seen, this pretrained model encounters challenges in generalizing to the specific FPAD task, given the substantial distribution shift between the WMCA domain and the ImageNet images that MSC uses to pretrain its model.
>
> The table also demonstrates the advantage of our simple method (without additional large-scale dataset) in the case of unseen attacks, which can be considered a form of distribution shift.
>
>
> Moreover, one motivation that can favor models without large-scale dataset for pretraining is dataset licensing, especially in the context of industrial applications like FPAD. Indeed, some collected data might not be licensed to be used in a commercial application (e.g. ImageNet21K).
>
> **Question. The memory learning in eqs. 2-4 involves ground truth that could be either  for anomaly or  for normal sample, but in sec. 3.2 there is a one-class AD objective, i.e., eq. 6, where no training anomaly examples are supposed to be available, so I'm confused that how the one-class AD model can be trained together with eqs. 2-4.**
>
> **Reply:**
> First, we note that in the first learning stage (Fig. 2a), anomalies are only used to further enrich the contrastive learning scheme (learning the representation z) but not for the memory learning (Mem). Indeed, at any time only normal prototypes are learned.
>
> The provided equations 2-4 present our framework of representation learning for unified anomaly detection, meaning it can work with or without (special case where =0) anomalies. Nothing circumvents the omission of anomalous samples in the first stage since it is driven by **unsupervised** contrastive learning. We can notice that the only use of the label y_k during the first stage is inside the memory layer Mem(x,y) that act as a pass-through if y=0, i.e. an anomaly.
>
> We made this point more clear in Sec.3.1 of the revised paper.

---

> > ### Author Response · Authors · 2023-11-22
> > **Reply to Reviewer 9JQe (2/3)**
> >
> > **Question. The concept of outlier exposure (OE) is defined and used in OOD detection for using external datasets as pseudo OOD examples to train OOD detection models, but in this work it seems it treats real anomaly examples as OE examples. Rather than using the so-called AD-OE concept, it may be clearer to use semi-supervised AD or open-set AD as in [3], [4], or [6]**
> >
> > **Reply:**
> > Thank you for your question. We made these points clearer in the revised paper.
> > From the perspective of availability of labeled data during training, semi-supervised learning should incorporate both unlabeled and labeled samples. For instance, [3] involves all unlabeled samples, labeled normal samples, and labeled anomalous samples. This is very different from our setting where we use labeled inlier and outlier samples.
> >
> > In [4], “open-set supervised AD” refers to the setting where during the training phase, the labeled samples (normal and anomalous) are used (thus, ‘supervised) and during inference, the model can detect both seen and unseen anomalies. From this perspective, for us, that setting “open-set supervised” is closer to OC than OE (in OC, all anomalous samples are unseen).
> >
> > While arguing that the OE is originally used in OOD detection, we believe that among these mentioned terms, OE-AD is still the most adapted. In other words, we embrace the OC term. Our OE-AD refers to training data that includes both inlier and outlier samples. In the revised paper, we made these points clearer.
> >
> > **Question. It is unclear why we need a method that works well for both anomaly detection and OOD detection, given the fact that the two tasks have quite different application settings.**
> >
> > **Reply:**
> > We would highlight that AnoMem is both unified and generic.
> > - In terms of its unified property, specifically regarding the availability of anomaly samples during training, AnoMem excels in both one-class and imbalanced outlier-exposure settings—a crucial practical feature in real-world scenarios where normal samples are initially available, and anomaly samples are acquired later, a scenario not adequately addressed by existing literature.
> > - Moving on to its generic property, AnoMem demonstrates effectiveness in object-/semantic-level anomaly detection, out-of-distribution (OOD) detection, and also in industrial applications like fine-grained anomaly detection (FPAD). In this real-world application, existing anomaly detectors and OOD detectors fail to work.
> > - While not the primary focus of our paper, we further demonstrate that our learning method can enhance the more general task of image classification. This aspect might have been overlooked in the initial submission, where different results for distinct tasks were presented in the same Table 4. To provide a clearer depiction of its capabilities, we have now separated the results for different tasks into Tables 4 and 5.
> >
> > **Question. Recent SOTA semi-supervised/open-set anomaly detection methods should be used in the experiment comparison to justify the advantages the work has, see [4]**
> >
> > **Reply:**
> > First, as mentioned earlier, our setting is different from your mentioned ones (i.e., not open-set supervised). Second, several recent methods including your mentioned model aims to solve anomaly localization rather than object-/semantic- level anomaly detection as in our work.
> > Finally, we note that the datasets considered in those anomaly localization papers are generally much smaller than our considered datasets (~5K images compared to ~100K with our datasets).
> >
> > **Question. Experiments on large-scale high-resolution image datasets, e.g. the popular setting that uses ImageNet-1k as the ID dataset, are missing for the OOD detection task.**
> >
> > **Reply:**
> > Thank you for your question.
> >
> > To the best of our knowledge, ImageNet-1k has not been utilized as the ID dataset in popular detectors, possibly due to its size, which may render it impractical for such purposes. For example, detectors like CSI [1] and MSC [2] have presented their results using CIFAR-10 as the ID dataset. CSI used a significantly smaller subset of ImageNet-1K, namely ImageNet-30, as their ID dataset. It is important to note that in the original paper, CSI did not provide results on CUB, and our results outperform theirs by nearly 30 points (the results of CSI on CUB comes from [5]).
> >
> > We would like to emphasize that we have already presented high-quality results for anomaly detection (in two settings), out-of-distribution detection, and fine-grained anomaly detection. We have chosen to leave the evaluation on other datasets/tasks for future work, as indicated in the paper.

---

> > > ### Author Response · Authors · 2023-11-22
> > > **Reply to Reviewer 9JQe (3/3)**
> > >
> > > **References:**
> > >
> > >
> > > [1] Tack et al., “CSI: Novelty detection via contrastive learning on distributionally shifted instances”, NeurIPS 2020.
> > >
> > > [2] Tal Reiss and Yedid Hoshen, “Mean-shifted contrastive loss for anomaly detection”. AAAI 2023
> > >
> > > [3] Ruff et al. “Deep Semi-Supervised Anomaly Detection”. ICLR 2020
> > >
> > > [4] Ding et al. “Catching both gray and black swans: Open-set supervised anomaly detection”. CVPR 2022
> > >
> > > [5] Mirzaei et al. “Fake it till you make it: Near-distribution novelty detection by score-based generative models”. ICLR 2023
> > >
> > > [6] Acsintoae et al. "Ubnormal: New benchmark for supervised open-set video anomaly detection". CVPR 2022

---

### Author Response · Authors · 2023-11-22
**General responses**

Dear PCs, ACs, and all reviewers,

We appreciate the valuable comments from the three reviewers that have significantly enhanced our paper. The previous version may have had some ambiguities in certain details. In response, we have revised the paper according to their feedback, marking the changes in magenta. Additionally, we've highlighted in brown the text that was either part of the initial submission which may have been overlooked by the reviewers.


Reviewers have raised concerns regarding **1)** the positioning of our model and **2)** additional experiments

1) We would like to emphasize the novelties and positioning of AnoMem as follows:
AnoMem stands out as the first to be both unified and generic.
    - In terms of its unified property, specifically regarding the availability of anomaly samples during training, AnoMem excels in both one-class and imbalanced outlier-exposure settings—a crucial practical feature in real-world scenarios where normal samples are initially available, and anomaly samples are acquired later, a scenario not adequately addressed by existing literature.
    -	Moving on to its generic property, AnoMem demonstrates effectiveness in object-/semantic-level anomaly detection, out-of-distribution (OOD) detection, and also in industrial applications like fine-grained anomaly detection (FPAD).
    - While not the primary focus of our paper, we further demonstrate that our learning method can enhance the more general task of image classification. This aspect might have been overlooked in the initial submission, where different results for distinct tasks were presented in the same Table 4. To provide a clearer depiction of its capabilities, we have now separated the results for different tasks into Tables 4 and 5.

2) Following the reviewers’ suggestions, we have now included several results of CSI and MSC. Once again, AnoMem outperforms SOTA methods like CSI and MSC in the industrial applications of anomaly detection and OOD detection, i.e., FPAD. It is worth noting that MSC relies on a pretrained-model of 14M images and using certain large-scale datasets may not be permitted in industrial applications due to restrictive licenses.

We are enthusiastic about continuing the discussion.

---

### Meta-Review · Area_Chair_nUXZ · 2023-12-07

**Metareview:**

This work introduces an algorithm for simultaneously handling both OOD and anomaly detection (AD) tasks. It is based on a combination of contrastive learning and a novel multi-scale memory bank implementation. Reviewers appreciated the novel and interesting algorithm, but thought that the idea of unifying the OOD and AD settings was not well-motivated. There were also concerns on the lack of evaluation of more recent methods (most methods compared are 2-3 years old) and on standard benchmark datasets like MVTec and VisA. Finally, there are issues with the presentation as detailed in the comments. The authors provided comprehensive responses that addressed some of these issues and one reviewer raised their score. However, in the following discussion, reviewers agreed that fully addressing the issues raised would require a significant rewrite (e.g. to properly motivate the need for a unified algorithm) and additional experiments (e.g. on recent baselines, "non-unified" baselines, and common benchmark datasets like MVTec and VisA). The AC agrees that the current version is not quite ready for publication and recommends rejecting the paper. That said, the AC thinks that there are good contributions in the paper and encourage the authors to revise it accordingly for re-submission.

**Justification For Why Not Higher Score:**

- Motivation for unifying OOD and AD settings is not quite clear
- Lack of comparison to recent baselines and on common benchmark datasets

**Justification For Why Not Lower Score:**

N/A

---

### Decision · Program_Chairs · 2024-01-16

Reject